# An Intergenerational Exploration of Discipline, Attachment, and Black Mother–Daughter Relationships Across the Lifespan

**DOI:** 10.3390/bs15070887

**Published:** 2025-06-30

**Authors:** Seanna Leath, Lamont Bryant, Khrystal Johnson, Jessica Bernice Pitts, Titilope Omole, Sheretta T. Butler-Barnes

**Affiliations:** 1Psychological and Brain Sciences Department, Washington University in St. Louis, St. Louis, MO 63130, USA; j.khrystal@wustl.edu (K.J.); sbarnes22@wustl.edu (S.T.B.-B.); 2Department of Psychology, University of Virginia, P.O. Box 400224, Charlottesville, VA 22911, USA; 3Department of Psychology, University of Michigan, Ann Arbor, MI 48104, USA; jbpitts@umich.edu

**Keywords:** Black mother–daughter relationships, discipline, attachment styles, intergenerational processes

## Abstract

Discipline is a significant predictor of parent–child attachment and relationship quality across the lifespan. Yet, much of the research on Black families’ disciplinary strategies uses a deficit and myopic lens that focuses on punitive punishment styles (e.g., spanking or taking away privileges). In the current exploratory qualitative study, we used an intergenerational narrative lens and thematic analysis to explore semi-structured interview data from 31 Black mothers (25–60 years, M_age_ = 46) in the United States around their mothers’ disciplinary practices during their childhood. We explored the connections that Black adult daughters made between their childhood disciplinary practices and their current disciplinary practices with their children, as well as their current relationships with their mothers. Adult daughters varied widely in their retrospective accounts of their mothers’ disciplinary strategies, which we categorized into three themes: (a) punitive, (b) logical, and (c) natural. We also identified three themes around how their mothers’ practices informed their current disciplinary practices with their own children: namely, (d) continuity, (e) mix, and (f) shift. Finally, we identified three themes around the current nature of their mother–daughter relationships: (g) strained, (h) progressing, and (i) healthy. The results highlighted the personal and cultural factors that informed Black women’s disciplinary strategies across two generations of mothers and revealed that when adult daughters shifted away from what they experienced during childhood—it was often towards less punitive strategies. Our exploratory findings also pointed to patterns regarding the extent to which Black adult daughters felt connected, validated, and supported by their mothers. The findings lend insight into Black mother–daughter relationship dynamics, particularly around the importance of communication patterns and emotional connection in the culture of discipline within families.

## 1. Introduction

“Black children raised without whuppings and fear can flourish. Rooting out violence in all forms—from our families, schools, and communities—is an essential step to challenging racist devaluation.” ([67])

Several scholars have documented disciplinary beliefs and practices[note 1] among Black families ([17]; [39]; [41]; [70]), and Black mothers are often a primary source of discipline and guidance for their daughters ([1]; [44]; [49]). Many early studies on disciplinary practices focused on lower-income Black mothers ([28]; [44]) and conveyed an underlying assumption that harsh disciplinary practices were endemic to this community of parents because mothers were dealing with excessive economic and sociocultural stress that undermined their parenting capacity ([43]). Yet, other research in this area indicates considerable diversity in Black mothers’ disciplinary practices ([23]; [57]; [89]), including a stronger endorsement of methods such as privilege removal (e.g., grounding a child or taking away a toy) rather than physical punishment ([1]; [31]). Furthermore, contextual factors, such as perceived social support ([43]), maternal characteristics (education level and socioeconomic status; [39]), and environmental concerns (neighborhood violence; [61]) are often associated with Black mothers’ disciplinary choices. Still, maternal discipline practices is an understudied area of research, particularly in thinking about the intergenerational nature of family dynamics and attachment patterns between Black mothers and daughters.

For instance, collective findings in the literature on Black maternal discipline ([8]; [39]; [41]; [49]), in conjunction with evidence on the transnational and ever-evolving nature of family relationships ([14]; [62]; [81]) suggest that it is important to consider Black daughters’ perceptions of their mother’s behaviors, as well. In thinking about Black mother–daughter dynamics, specifically, mothers are often salient role models for Black girls as they develop their self-concepts and belief systems about the world ([8]; [34]; [48]; [82]; [83]; [84]). Black mothers may try to use discipline to teach their daughters lessons they believe will serve them well in adulthood; yet, we know little about how adult daughters understand and perceive their mother’s choices around discipline as a form of guidance and support. In the current study, we utilize intergenerational narrative theory ([59]; [54]) and in-depth interview data to explore how Black adult daughters make sense of the maternal discipline they experienced during childhood, with a particular focus on how their meaning-making around these experiences informed their current disciplinary practices with their children as well as their sense of attachment in the current relationships they have with their mothers in adulthood.

## 2. Theoretical Framework: Intergenerational Narratives Among Black Families

Individuals make sense of their lives through narratives ([37]; [45]). The stories we tell ourselves simultaneously shape how we understand our lived experiences ([35]; [51]) and structure what and how we share our experiences with others ([60]; [56]). Personal stories help individuals bridge the past, present, and future into a cohesive narrative identity that offers a sense of purpose and direction in how they move through the world ([53]). As captured within the African concept of ubuntu, “I am because we are”, individual’s life narratives are connected and constructed through other people and surrounding ecological contexts ([19]). We learn more about ourselves through our relationships with other people as well as the socialization we receive about social and cultural norms and expectations ([66]; [78]). In the present study, we use intergenerational narrative theory ([37]; [59]) to consider Black adult daughters’ meaning-making processes about their mothers’ disciplinary practices and its subsequent role on their own discipline norms. Scholars suggest that mothers’ beliefs and behaviors are a strong predictor for their children’s beliefs and behaviors ([8]; [9]; [34]; [68]). While most of this work focuses on early childhood development, the notion that values and practices are passed down from mothers to their children ([21]; [65]; [81]) likely carries forward into adulthood, as well. Yet, questions of intergenerational continuities in discipline, or whether maternal discipline and child-rearing inform the development of their adult daughters’ beliefs about motherhood and discipline, remain largely unexplored.

Thus, we use an intergenerational narrative lens because it allows us to explore Black women’s autobiographical meaning-making across the developmental timeline of childhood to motherhood. [37] ([37]) framework draws upon ecological systems theories ([15]; [33]; [86]), which highlight how individuals develop within multiple interacting systems or “narrative ecologies” (e.g., families, schools, and broader communities). Individuals produce interpretations of their experiences in home and community settings that inform their understanding of self (i.e., “What does this experience say about me?”) ([55]). As Black women transition into motherhood, their narrative identities of self—“Who am I? Who am I as a mother?”—may become salient in new ways as they negotiate the intersecting demands of personal health, careers and professional goals, friendships and intimate partnerships, and caretaking roles ([47]; [58]; [76]). Women often learn their beliefs and values around motherhood, including what it means to be a ‘good mother’, through behaviors that were implicitly or explicitly modeled in their families as well as what they witnessed in community settings like church and social media ([3]; [11]; [65]; [72]; [75]).

While multiple narrative ecologies can inform Black women’s beliefs and behaviors as they transition into motherhood, they may rely more heavily on prior experiences and conversations with their mothers to help guide their parental decision-making processes ([8]; [21]; [29]). Parental decision-making includes a range of things, such as making choices for a child’s well-being in housing, education and schooling, health care, and recreational activities. Scholars suggest that as new mothers make myriad decisions for their children during this phase of life, their mothers’ prior and current support and guidance plays a key role ([38]; [16]; [64]). Still, we know little about how family dynamic norms and the nature of Black women’s parent–child relationships with their mothers (i.e., communication patterns, relational closeness, and conflict resolution (or lack thereof) informs the support they solicit and receive from their mothers in adulthood ([81]). Consistent with research on intergenerational processes and narrative identity ([45]; [59]), reflecting on their childhood experiences as they transition or mature in their motherhood journey may offer unique opportunities for Black adult daughters to evaluate their mothers’ decisions around discipline and consider what they want to maintain as well as the areas they want to change. By using an intergenerational narrative lens, we can explore maternal discipline as an evolving component of Black women’s life stories, particularly in how they reconstruct their childhood past to embody and embrace the type of mother they would like to be for their children in the present.

## 3. Maternal Discipline and Black Mother–Daughter Attachment

To date, one of the most influential typologies of parenting discipline is based on the work of [7] ([7]), who focused on levels of warmth and control in how parents responded to their children. Baumrind characterized three main types of parenting approaches, including (a) authoritarian parents (i.e., low in warmth and high in control), (b) authoritative parents (i.e., high in warmth and high in control), and (c) permissive parents (i.e., high in warmth and low in control). In relation to discipline, Baumrind stated that authoritarian parents tended to expect their children to obey their rules and expectations without explanation, authoritative parents tended to explain the reasoning and logic behind family rules but maintain the expectation that children adhere to their roles, and permissive parents tended to avoid rules or consistent boundary setting with their children.

Although disciplinary practices (e.g., spanking, yelling, time-outs, talking through consequences, etc.) do not necessarily map directly onto Baumrind’s broader typologies of parenting, scholars have found significant associations among parenting typologies, disciplinary practices, and children’s attachment patterns ([24]; [25]; [69]). Specifically, according to attachment theory, children who consistently experience responsive and sensitive caregiving develop an expectation that others will be supportive and available in times of need ([80]). Responsive and sensitive caregiving includes mothers’ disciplinary practices, such that mothers who de-emphasize children’s autonomy or agency to exert control over their children may weaken or harm a child’s capacity for secure attachment ([26]; [41]; [42]; [46]; [50]).

Attachment scholars have also drawn attention to the unique experiences of Black families, including the extent to which discipline and attachment practices are sustained across generations ([6]; [18]; [40]; [80]). How Black mothers choose to correct and guide their daughters—under the umbrella of ‘discipline’—informs the sense of trust and connection within the relationship. Relational patterns of discipline and attachment are established and reinforced throughout childhood and adolescence ([24]; [44]; [67]; [70]), and results suggest that a foundation of the mother–child relationship is the tenuous balance between a mother’s role of providing nurturance (i.e., physical, emotional, and instrumental support) and providing discipline as a form of structure and guidance. To date, most of the research on these typologies and parent–child attachment has focused on the childhood years. Yet, the way mothers respond and attune themselves to their children is a cornerstone of this relationship dynamic across the lifespan ([27]; [46]; [68]; [77]), even if the need for ‘discipline’ shifts over time.

In the current paper, we aim to explore ‘becoming a mother’ as one of those critical phases of development, wherein Black adult daughters may develop new understandings and perspectives on the disciplinary choices their mothers made while raising them. In reviewing the literature, we did not locate any studies where adult daughters reflected on how their mother’s disciplinary behaviors in the past and present carried over into the relationship they have with their mother in the current period. Our study was guided by three primary questions: (1)What types of discipline and consequences do Black mothers recall from their childhood?; (2) How do Black mothers make sense of the disciplinary experiences they recall from childhood—as a child and as an adult mother with children?; and (3) In what ways do Black mothers draw connections between their disciplinary experiences as a child and their current relationship with their mothers? We aim to advance how we think about Black mother–daughter relationships across the lifespan, with a particular focus on intergenerational disciplinary practices and parental attachment.

## 4. Method

### 4.1. Participants

The current sample includes interview data from 31 Black cisgender mothers (25–60 years, M_age_ = 46) in the United States. These data were from a larger mixed-methods study on how Black mother’s race-related beliefs and experiences inform their parenting practices. They lived across the United States, including the Southeast (n = 12, 39%), Midwest (n = 9, 29%), Southwest (n = 4, 12%), Northeast (n = 3, 10%), and West (n = 3, 10%). The majority were African American (n = 23, 74%), and the sample also included mothers who identified as African (n = 2, 6%), Caribbean (n = 2, 6%), or bi- or multiracial (n = 4, 14%). Most of the women had a graduate degree (i.e., Master’s, PhD, MD, or JD; n = 18, 58%) or a bachelor’s degree (n = 8, 26%); three women (10%) reported some college or trade school and two women (6%) did not indicate their highest educational attainment. They were in a range of occupational fields, including education, scientific research, medicine, social work, mental health consulting, and nonprofit leadership. At the time of data collection, the women had between one to six children, ranging in age from three months to 21 years old. Twenty-two (71%) of the women were partnered or married and most of these relationships were with Black men (95%). See Table 1 for specific demographic information on each mother, including the discipline they discussed and their current characterization of their relationship with their mothers.

### 4.2. Procedures

After receiving university IRB approval (study #202303008), the primary investigator (PI) recruited participants through a combination of social media recruitment and snowball sampling (i.e., PI asked mothers who participated to send the recruitment flier to other eligible mothers). Eligibility criteria included being a Black or African American mother currently residing in the United States and raising at least one Black child (biological, foster, or adopted) at the time of data collection. For social media recruitment, the PI posted the flier with study details on Facebook, Twitter, and Instagram. For snowball recruitment, the PI emailed a copy of the flier to mothers who were willing to send it to other eligible Black women. Interested women could sign up for an interview through Calendly, an online scheduling service. The PI emailed the informed consent and demographic sheet to potential participants after they signed up for an interview. After receiving the informed consent and demographic information back and confirming eligibility, the PI emailed a Zoom link for the interview. The interview team included two Black women (PI and a doctoral student) and one Latina woman (doctoral student) with expertise in child development, family system theories, and qualitative methodologies. The one-on-one interviews ranged from 50 min to 120 min (*M* = 70 min); the research team kept their cameras on for the duration of the interview and most of the mothers did, as well. The audio files were sent out for professional human transcription to Rev. Once returned, all names and identifying details were changed to protect the confidentiality of the mothers and their families. The transcribed interviews were then reviewed by the PI for accuracy and uploaded to a secure university data server.

### 4.3. Interview Protocol

The semi-structured interview guide had four main sections: disciplinary experiences during childhood (e.g., From what you recall, what types of disciplinary styles did your mother use when you were younger?), current parenting practices (e.g., What does conscious parenting mean to you?), race-related beliefs and experiences (e.g., In the age of social media, we are exposed to frequent stories of police brutality and violence against Black people and communities. To what extent do these stories influence you as a mother?), and advice to other parents (e.g., What do you wish you could tell yourself when you were just starting your parenting journey?). For the current study, we focused on the mother’s responses in the first two sections of the interview on the discipline they experienced as a child and their current disciplinary practices with their children. These questions included, “Let us start by talking about your own relationship with your mother. How would you describe your relationship?,” “What did you think about your mother’s disciplinary style when you were a child?,” and “What do you think about your mother’s disciplinary style now, as a mother with children?” The interview protocol is attached as a Appendix A and available upon request from the PI.

### 4.4. Coding Analysis

We used inductive thematic analysis ([13]) to analyze the interview data. Inductive thematic analysis is a flexible qualitative approach that scholars can use to identify patterns within and across participants’ narratives regarding their thoughts, feelings, and beliefs about their prior disciplinary experiences and their current disciplinary practices as a mother. An important element of inductive qualitative research is to consider the scholarly positionalities and areas of expertise among the research team because our backgrounds, experiences, and beliefs can influence the interpretation of the data and the identified themes ([13]; [73]). The coding team involved the PI and first author (a Black woman), a Black female undergraduate student, and an external auditor (Black female graduate student). Furthermore, the authorship team involved six Black gender-diverse scholars from a range of social classes (working class to upper-middle class based on educational attainment and household income) and professional backgrounds (undergraduate students to a full professor). Collectively, our research interests include identity development, Black family socialization processes, educational access for Black students, social support processes for Black queer youth, health and wellness among Black artists, and stress and coping processes in Black families. In thinking about our positionalities in relation to the current study, we are committed to amplifying strengths-based scholarship on Black family processes. There were three main phases of data analysis after the pre-coding process (i.e., selecting and finalizing excerpts): (1) identifying preliminary themes through note taking, (2) refining the themes and developing the codebook, and (3) finalizing the codebook and summarizing the narrative findings.

Before we started the analysis process, the PI and her student compiled all excerpts into an excel file that had eight columns (i.e., pseudonym, participant ID number, (RQ1) childhood discipline and notes, (RQ2) disciplinary practices with their children and notes, and (RQ3) current relationship with their mother and notes). After extracting excerpts for each of the questions, we transitioned to the first phase of coding analysis. To identify preliminary themes, the PI and her student independently reviewed the excerpts and added notes for each research question. These notes included broad and semantic descriptions, such as (RQ1)—physical punishment and yelling or (RQ3)—feels like mother is a friend and trusted advisor. At this stage, we focused on participants’ explicit descriptions to get a sense of the data. After completing this process for all three research questions, the PI met with the external auditor to review the notes and begin clarifying themes and definitions. We compiled all notes into a separate tab and consolidated them into thematic categories with the goal of moving from basic descriptions to more abstract interpretative themes. For instance, for RQ1, we designated semantic notes about “spanking, slapping, choking, objects being thrown, cursing, name calling, threats of bodily harm, and yelling” as punitive disciplinary consequences. For RQ2, we translated notes like “can appreciate the whoopings her mother gave, though it is not the participant’s parenting styling” as a shift in their current disciplinary practices as a mother. We reviewed family studies and discipline literature ([12]; [27]) to name and define some coding categories. Final themes and definitions are available in Table 2.

After refining themes and creating a codebook with the auditor, the PI and student coder met weekly to assign codes. To establish intercoder reliability ([63]), we worked through the first third of the data (*n* = 10 excerpts for each research question) and alternated who led the discussions for their applied codes. Thus, for example, the PI led the coding discussions for the excerpts with RQ1 and her student led the coding discussions for RQ2. We established 88% intercoder reliability on these excerpts and then assigned independent codes for the remaining data. The second intercoder reliability check was 85%, and as a final reliability measure, we had a full team meeting with the PI, student coder, and auditor to discuss coding assignment discrepancies. The primary coding discrepancies involved characterizations of the adult daughters’ current relationships with their mothers, especially regarding healthy versus progressing relationships (e.g., if a woman mentioned conflicts with her mother but actively talking through those conflicts, does that quality as healthy or progressing?). To resolve these discrepancies, we returned to the original interview transcripts and the women’s explicit descriptions of their relationships with their mothers. After coding, the team engaged in an open discussion about overall patterns in the data and generated individual summaries of the findings.

## 5. Results

In the current study, we explored retrospective maternal disciplinary norms that Black adult daughters recalled from their childhoods. We were interested in considering mothers’ perspectives on how their maternal childhood disciplinary experiences informed their current disciplinary practices as a mother as well as their current relationships with their mothers. While we recognize that a range of factors inform parent–child relationships across the lifespan, we aimed to explore discipline as a context for attachment between Black mothers and daughters. Below, we summarize and review findings from the three primary research questions and offer representative examples. For definitions and additional examples of disciplinary categories (RQ2), see Table 2.

### 5.1. RQ1—Disciplinary Experiences During Childhood

We found three main characterizations of discipline in response to our first research question about the consequences mothers recalled from their childhood. Specifically, mothers described punitive (i.e., a punishment rooted in negative reinforcement that is doled out by an adult in response to a child’s action or non-action), logical (i.e., a consequence that is imposed by an adult and related to a child’s perceived misbehavior), and natural (i.e., a consequence that is not imposed by an adult, but occurs in response to a child’s behavior or choices) consequences. Each type of discipline category included a range of parental actions. For instance, punitive consequences included spanking, slapping, throwing objects, invoking guilt to change their behavior, using expletives or name calling, yelling, ignoring a child (i.e., the silent treatment) and dismissing their daughter’s feelings, concerns, and attempts at explaining their behavior. Logical consequences included experiences like a mother taking away their daughter’s game system after they continued to play it past curfew or a mother telling the daughter that she has to do her homework right after school after she forgot to do it several times in a row when she was allowed to set her own ‘homework time.’ Although logical consequences were sometimes punitive in nature (e.g., taking away something the daughter enjoyed), they were always tied to a perceived rule violation. Finally, natural consequences included things like their daughter being cold when they went outside because they refused to wear a coat or a daughter being tired at school the next day after they stayed up past their bedtime.

With both logical and natural consequences, mothers tended to talk with their daughters about their behaviors to make connections between their choices and the subsequent disciplinary consequences. This type of communication was less common with punitive discipline practices. In terms of each category, punitive consequences were the most common category, with 23 (74%) of the women recalling at least a few instances of punitive discipline. Eighteen (58%) women discussed the logical consequences their mothers imposed, and three (10%) women described natural consequences. Many mothers (*n* = 15, 48%) recalled a mixture of at least two discipline categories (e.g., punitive and logical), and sometimes, they noted shifts in their mother’s discipline based on their age (e.g., fewer punitive consequences or less physical discipline as they aged). We recognize that we used retrospective accounts from the women on their childhood experiences. Thus, it is likely that the women experienced forms of discipline that they do not recall or that they opted not to share. On the other hand, their interview responses likely reflect the most salient disciplinary practices they remember, and they involve the practices that the mothers directly connected to their own parenting practices and their current relationships with their mothers. See Table 2 for definitions and additional examples of each discipline category.

### 5.2. RQ2—Disciplinary Reflections and Current Practices as a Mother

Within the second research question, we explored Black adult daughters’ current disciplinary practices in relation to what they recalled from childhood with their own mothers. Specifically, we asked the women to reflect on whether they used similar disciplinary practices with their own children as their mothers had used with them. Based on their descriptions, we identified three thematic categories: continuity, mix, and shift.

**Continuity.** Within the continuity category (*n* = 5, 16%), mothers stated they kept many of their disciplinary practices the same. These adult daughters pointed out the positive attributes they perceived from their mother’s disciplinary practices as they made connections to their own approaches. An important element of the continuity category was that they believed their mother’s approach was an effective way to raise children, and they elected to follow suit in their disciplinary philosophy as a mother with their children. The mothers in this category reported a combination of punitive, logical, and natural disciplinary consequences, and most of the punitive consequences involved restrictions versus physical punishment. Florynce, a 33-year-old African American mother with one child, shared,

We were never spanked. We did a lot of talking. A lot of conversations. The thing that I did get grounded for though, would be grades. So, like in eighth grade for example I got three B’s on my report card and I was grounded for six weeks. And so, I was pissed. As a kid, I was pissed. And now we’ve talked about it as an adult and she feels bad. And for me, now, my ultimate goal and what I hope [with my kids] is to keep our relationship and connection, because I think, especially now with two tweens as they’re coming into adolescence, I want there to be communication. And they’ll mess up or whatever, but we can work through that. And I take a lot of this from my mom, for sure.

Like Florynce, most women who discussed continuity with their mother’s disciplinary practices recalled logical consequences and conversations with their mothers about their behaviors. These logical consequences often involved a mixture of restricting treasured items or activities (e.g., preventing them from seeing their friends or taking away television privileges). Furthermore, when the women discussed their feelings of anger or frustration as a child with some of their mother’s disciplinary practices, as seen in Florynce’s case, they also recounted the redeeming or useful aspects of their mother’s approach. For Florynce, that involved the open and ongoing conversations she had with her mother as she moved through adolescence and young adulthood. Relatedly, Jada, a 26-year-old Ghanaian mother with two children, said,

Honestly, I think she was very effective because there was a lot of communication and there were a lot of warnings before steps were taken to other types of discipline, or any type of physical punishment or handwriting that we had to do. It was always a warning. And it was explained to you and so I understood it. I saw other friends whose parents were not like that. They would just physically discipline them in front of anybody. So my brothers and I were really fortunate because my parents did not do that. My mother did not believe in humiliating us. So my parents had a different disciplinary style from most other Ghanaian parents. We were lucky because they never humiliated us. That was not part of the agenda. It was usually, let me talk to you privately. I knew, as a mom, I wanted to model a lot of what my parents did with me. I wanted to model that and I work in the education field, so you see quite a bit and I wanted, my children should be rooted in culture and respect and proper etiquette.

Like Jada, other women in the sample compared their disciplinary experiences during childhood to what their friends encountered at home. By doing so, this seemed to help them put their mother’s practices in perspective and provide a point of reference to think about whether what they experienced was normative at the time. In talking about her mother’s approach, Jada referenced her cultural identity as a Ghanaian American to note her parents’ departure from what she perceived as a common disciplinary practice within that community. She believed her mother’s approach stood out compared to that of other Ghanaian mothers around her because she talked with her children about their wrongdoing and provided warnings about their misbehaviors. The three women who talked about receiving natural consequences offered similar reflections. For example, Claudette, a 41-year-old, African American woman with three children, said,

I think it was revolutionary, honestly. When I talk to my grandparents or older aunts and uncles, it stands out to them how my mom parented us, because it was not how they were brought up and not how they brought up their kids. So we always stood out. My mother got ridiculed for how she disciplined us, because it wasn’t the way everyone was doing it. And in leaving home and realizing, “Wow, she was telling the truth.” That kind of thing. And then it was also meeting other people and realizing, “Your mother said that” or “Your mother did that?” and realizing the blessing that I had. People telling me about getting spankings or whippings or beatings, and that’s when I realized, “Wow, my parents really did things differently.” And of course the icing on the cake was having my own kids. That was definitely the thing that helped me really view her in a different way.

Claudette was one of the few women who mentioned that her mother used a combination of logical and natural consequences, and she stated that her mother’s methods were very different from how other adults in her family approached child rearing at the time. She expressed appreciation for her mother’s steadfastness and willingness to be different, and she used many of the same tactics with her own children. Overall, the women continued maternal disciplinary practices that they perceived as fair, effective, and aligned with their parenting values.

**Mix.** In the second theme, six adult daughters (19%) discussed the disciplinary strategies they used that were like their mother’s approaches, as well as the practices they adopted that were dissimilar. As Sojourna, a 33-year-old biracial mother with two children, said,

They [parents] did instill some great values. I think professionalism, timeliness, and those kinds of things were instilled that were good, but there are some other things such as, the fear, the spanking and all that stuff that I’d prefer not to do. With my parents, it was more through fear that you listened. As a kid, it was always in the back of my mind, I’m like, “Wait, I have questions about this.” I want to understand and know why I’m doing this and that. But it was considered rude to even ask why. Which I didn’t like very much. So I think there was some feeling of resentment. So with my kids—in order for them to respect me enough, to listen to what I have to say, not that they have to do what I say, but in order for them to respect me enough, to at least listen and consider the guidelines that I’m setting forth, I need to be willing to listen to the things that are important to them.

Like Sojourna, most of the mothers described eliminating or reducing the types of punitive discipline they experienced with their mothers during childhood but retaining many of the family values around discipline that were consistent with how they wanted to guide their children. Daisy, a 40-year-old African American mother with one child, shared,

My father was very strict, with a lot of rules. A lot of rules that didn’t seem to make sense. Rules just for the sake of rules. But not rules that were intentional or really even applicable to the individual child. And definitely physical discipline. Even for my mother, it was like we were all kind of in a little prison. Dealing with a warden [her father]. So there are things that I definitely still took from my mom and there are things that I threw away. I look at it now in terms of who built the relationship and who didn’t. Who was invested in the relationship with us, in their children, versus who wasn’t. When I look at what parent has the relationship with their children now, it’s my mother, and who was tolerated was my father. Just like, I knew I could tell my mother anything, even if she didn’t want to hear it, I could tell her anything. And I want my kids to have the same thing with me. But when I think about discipline, I knew I definitely didn’t want to use physical discipline. And I tend to lean more into it terms of planting the seeds.

There were two moms in the sample who described more permissive parenting practices from their mothers (i.e., discipline style where parents are lenient and often lack set rules or boundaries with their children; parents are often warm and nurturing and may be more like friends than authority figures). In these cases, the women kept the relational practices that facilitated closeness and connection with their children, but they enforced more boundaries and rules than what they had experienced during childhood. Shanice, a 35-year-old African American mother with two children, said,

I kind of wish she was a little stricter, as far as like, not allowing me to tell her to shut up. What else? I’m thinking about my husband’s, my mother, my in-laws, their disciplinary style and how they turned out, and certain things I do like about it. I’m a Christian and a lot of their disciplinary things [with my husband] connected to the Bible and we do this because God says blah, blah, blah. And my mom didn’t do that as much. I wish she would have made sure I was respecting her the way I would want to be respected as a mom.

As seen with Soujourna, Dasiy, and Shanice, the connecting thread across excerpts within the mixed theme involved how the women understood their mother’s disciplinary practices—as a child (past) and as a mother (present). Overall, women in the mixed category were able to understand why their mother disciplined them in the ways they did during their childhood, even if they did not agree with it. Thus, as a mother, they retained the elements of their mother’s beliefs and practices that aligned with their own values and perspectives, and they left the rest in the past.

**Shift.** Within the third theme, 20 adult daughters (65%) talked about how they intentionally diverged from what they experienced with their mothers during childhood. As noted by the frequency counts across the three themes, most mothers demarcated a significant shift in how they were raising their children compared to what they experienced during childhood. In most cases, these shifts involved using less punitive measures (e.g., spanking or yelling) and many adult daughters discussed generational differences in parenting practices and beliefs. For some mothers, this shift in discipline occurred in response to research they did around parenting. Sylvia, a 34-year-old African American mother with one child, shared,

Now, I’ve deviated. My spouse is more so the one who still says, “If we were spanking them, we wouldn’t deal with this.” But the more I got into my own consciousness around anti-violence, I was like how could I be anti-violent and be spanking my children? Plus, as a child I didn’t really…that didn’t really work for me, right? Because I always felt like if it worked, why do people get repeat spankings, right? If it really worked, you shouldn’t get spanked more. Coming into my own consciousness and then being introduced to Dr. Stacey Patton. Once I started getting some of her information, I was just like whoa, I’m done with this spanking crap, yeah. But my mom still holds her line. She’s like, “Yeah, well there were four of you guys.” My mom was a physical therapy aid and she was doing manual labor. She was lifting people out of whirlpools and she was like, “I’m tired by the time I get home I just needed you all to fall in line and spanking did it.”

As demonstrated in Sylvia’s excerpt, many of the mothers in the shift category reflected on the discipline they experienced as children and decided they wanted to do something else. In some cases, they had explicit conversations with their mothers about the underlying logic behind the discipline they received. This sometimes helped them contextualize their mother’s choices, even as they made different choices for their children. For several women, their mother’s choices related to the support and resources they had (or did not have) available to them, including access to parenting education or social support to help alleviate some of the caretaker stress. Gwendolyn, a 27-year-old African American mother with two children, voiced,

Honestly I decided a long time ago I wasn’t going to spank. And then I had to build up to say, “What am I going to do instead?” And honestly, the first time I held my baby and I was looking at her—I think I was changing her diaper and I looked at her little booty and I was like, “I don’t ever want to hit that.” It just kind of came over me and it was probably the hormones too, right? I was like, “I’m not going to do it.” I wish my mom had tapped into that more. It’s hard especially being that my family was very educated, right? Like my grandfather had a PhD and all these other things. So it wasn’t about degree and education—it’s really about a mindset. I do agree she did the best she was able to.

As seen with Gwendolyn, in many cases, the mothers believed their mothers were doing the best they could with the information and parenting skills they had at the time. This was significant regarding the connections they made between their childhood disciplinary experiences and their current relationship with their mother. Maya, a 46-year-old African American mother with one child, said,

I don’t fault my mother. I think that parenting changes with new information and different periods of time. I think she disciplined me in the way that was common for ‘80s kids. So it is what it is. We also didn’t usually use car seats. We use car seats now. I know some people that have really deep seated feelings about getting spankings as children, and I don’t. I definitely do think that I probably deserved some of those spankings, and even if I’ve chosen to do something different with my kids, I don’t fault my mom for using the information that she had available at the time to discipline me the best that she could.

Similar to Maya, other adult daughters noted their mothers were following common social and cultural norms around discipline when they were growing up. This seemed to alleviate the distress or frustration they might have had with their mothers and allowed them to maintain a close relationship with them in adulthood.

### 5.3. RQ3—Characterizations of Current Mother–Daughter Relationships

Finally, our third research question explored how the women described the nature of their mother–daughter relationships in adulthood. A multitude of factors inform mother–daughter relationships across the lifespan, so we focused on the interview protocol questions that asked them draw connections between their mother’s disciplinary styles and their sense of attachment and connection to them as adults. We also asked them about what they remembered thinking about their mother’s discipline—as a child—and what they thought about their mother’s practices now—as an adult daughter with children. We hoped these exploratory connections would offer insight into considering the developmental processes associated with discipline and relationship attachment among Black mothers and daughters across the lifespan. We identified three main categories of current relationships: strained, progressing, and healthy.

**Strained.** Within the first theme, seven adult daughters (23%) described relationships with their mothers that were characterized by conflict or distance and a general lack of closeness. Some daughters were not in contact with their mothers at the time of data collection, and they did not consider their mothers a primary source of support or a role model for how they were raising their children. Angela, a 41-year-old African American mother with three children, said,

Well my mom, she was a yeller and a cusser. And if you didn’t do something she wanted done, you were getting a whooping. You were scared all the time because you didn’t know what her reactions were going to be. It was annoying. It was scary at times. I think I had a lot of anxiety. Just not even knowing what was going to happen every day. I’m just doing everything possible not to be my mother and to make sure that my kids know that I’m there for them through everything. And I want them to understand that they can express themselves. They have an opinion. They can be heard. So they don’t go out here thinking that they’re not worthy of whatever it may be that they want in this world. I’m trying to do the exact opposite of what was done for me. And our relationship now—it’s strained. It’s strained. It’s a tough situation to deal with her. I can talk to her, but at the same time, she’s very difficult to deal with because I just feel like she’s very toxic and a lot of it comes from her upbringing and how she was dealt with as a child. And it’s a lot of things that she hasn’t dealt with and she tends to take it out on other people. She’s been through a lot. I understand that. But when you’re dealing with an adult child, I feel like there are certain ways that you should go about speaking to them. She is just not that person and she wants everything her way or it’s no way. So I distanced myself.

Although Angela maintained a semblance of a relationship with her mother, she believed that it was difficult to rely on her mother as a source of support and she felt upset about her mother’s poor communication patterns and emotional immaturity. In response, she set a personal boundary for herself and distanced herself. Many of these patterns seemed to stem back to the parent–child relationship Angela and her mother shared during childhood, where she received harsh disciplinary consequences and her mother struggled to provide a secure and safe home environment for her children. This was true for other women in the ‘strained’ relationship theme, as well, in that they often reported a period of distance or disconnection with their mothers. In a few cases, the strained nature of these relationships started in childhood and concurred with stretches of time when their mother was absent—either physically or emotionally. Maya, a 46-year-old African American mother with one child, remarked,

Okay, so I’ll say that I lived with my mother for only one or two years of my life, even though she was around. I lived mainly with my mother’s sister until I was about four years of age. And then my father and my stepmother and then my maternal grandmother. My mom was present during that time, but we didn’t live with her. And so as far as my relationship with her, she’s my mother. But when you think about a mother–daughter relationship, we don’t have that. I didn’t grow up crying on my mother’s shoulder. My mom was not the one who I ran to when something went wrong. As a matter of fact, my mom is very sensitive. So even though she can say whatever she wants to, you can’t say what you want to without her having an attitude. We went 10 years without speaking because of what she felt I said about her. So our relationship is more like friends.

As noted in both examples, strained mother–daughter relationships frequently entailed a lack of emotional support and frequent conflict between the two of them based on unmet needs. In some cases, the women mentioned that their mothers were overly critical or controlling during their childhood or that their personalities clashed or had challenging family dynamics (e.g., housing insecurity or intimate partner violence). Some women mentioned, as children, feeling stressed during interactions with their mothers and often expressing difficulty in talking about their emotions or other things that were important to them. In adulthood, they were able to make more intentional choices about how they engaged their mothers in ways that supported their well-being.

**Progressing.** In the second current mother–daughter relationship theme, 10 adult daughters (32%) discussed how their relationships with their mothers felt constrained by past or current conflict. Yet, to be coded in this category (rather than the strained theme), both parties were actively improving the sense of care and trust in the relationship. Generally, this involved the women’s mothers being accountable for past behaviors that left them feeling disconnected, insecure, or unheard. This accountability included their mothers being open to healthier boundaries and changing their communication and relational patterns in ways that felt more supportive to their daughters in adulthood compared to what was available to them in childhood. Imani, a 32-year-old African American mother with two children, stated,

My relationship with my mother is good. There was a barrier because she always wanted me to be more girly and I was always the tomboy and very comfortable being the tomboy, so we butted heads a lot on that. And it’s just recently that we’ve been able to have open discussions about it and realize where we both had mental blocks against it or still have issues with it. So, to hear some of the issues she has and for her to hear mine and for us to kind of come together and try to work through it, I feel closer to her now. Growing up, I didn’t really think she liked me, but it was always because…you know, it’s that parent/child relationship where neither one of you is really feeling heard or understood. But it’s recently shifted because I became a mom, and I had a failed marriage and I left my husband and literally had to move back in with her. So, us having to communicate when it comes to my son and life in general, it’s really become a bridge to be able to talk about things that we probably never talked about before.

Imani’s excerpt encapsulated the reflective process that mothers in the ‘progressing’ theme offered during their interviews. Specifically, she outlined some of the conflicts that she and her mother experienced during her childhood (i.e., clashing gender role and personality expectations) and then explained how they moved past this prior conflict after open and honest dialogue. Her words revealed the hurt she experienced as a child (e.g., “I didn’t really think she liked me”), and the ways the recent circumstances around her divorce impelled her and her mother to communicate about issues from the past. When describing the discipline she experienced from her mother during childhood, Imani discussed being frequently ‘grounded’ from going out with friends when her mother considered her too disrespectful or defiant. To Imani, she was often trying to communicate her needs as a child and her mother was unwilling to hear her out. Her mother’s willingness to listen actively and patiently in adulthood had improved their relationship considerably.

Many of the ‘progressing’ mother–daughter relationships included tension from unmet expectations; mothers tended to have future visions about their daughters that did not align with who their daughters were developing into as young women. For instance, Henrietta, a 33-year-old African American mother with six children, said, 

I’m pretty close to both, but I can relate more to my dad versus my mom because we did have that teenage rough patch I think everybody hits. And it took a long time to fix that teenage rough patch. She didn’t really yell or anything like that, it was just very…it was a nag. Kinda authoritative. My mom wanted me to follow this mold of going to school, having a career, and traveling, doing all of that. But she wanted me to do it her way versus my way, so we clashed and she would often take things away or refuse to talk with me. Now, it’s completely different. Now with my kids, they’re [parents] just like, oh come get all the sugar. They’re totally different as grandparents. And my mom and I have become closer once we got over the patch. When I first had my eldest daughter, she was kind of like, “Okay, well you should do it this way.” Almost like when I was a teenager. I had to, in the nicest way, be like—no. This is my baby—we’re going to do it this way. She would try to say her two cents and then she would be like—you know what, it’s your baby. You do it how you want. So that’s where we are now. We’re a lot closer now and she gets it.

As seen with Henrietta, the bridge for some mothers and daughters was related to the adult daughter’s parenting of their own children. When their adult daughters came to them as new moms who were trying to figure things out, they offered advice and affirming support that helped to mend or heal the prior divides and areas of tension within their relationship. This was especially true when mothers were willing to respect the choices their adult daughters were making as parents, even when these choices were different from what they had done with them while raising them. This type of respectful assurance helped many of the women feel closer to their mothers and seemed to be paving the way for healthier attachment patterns.

**Healthy.** Finally, in the third theme, 14 daughters (45%) stated their relationships with their mothers entailed open and honest communication, as well as a sense of love, affection, and consistency. The women used words and phrases, such as being “close” to their mothers, being able to “express themselves,” and knowing that their mothers could be “counted on” as sources of support. We also found that mothers might have experienced any of the three disciplinary styles (punitive, natural, or logical) when they were children and still report a healthy relationship dynamic with their mothers. In most cases, adult daughters who experienced punitive discipline reported a shift in their current mothering practices, but this shift did not mean that they did not share a close and warm relationship with their mother. As an example, Misty, a 47-year-old African American mother with two children, said,

I would say I had pretty close relationship with my mother growing up. I think she did a really good job of balancing the nurturing side of parenting with the other aspect of it, which is more of the disciplinarian side. So she was someone who growing up, we spent every evening together with my mom, all of us in the living room. We ate dinner together. We traveled with her, just quite close. That has continued throughout even my adulthood. I would say that we’re probably even closer now, and that’s only in part because my parents are now in their late 70s. I talk to my mom once a week. She lives in North Carolina and I live in Kentucky. I’m also the youngest of four girls, and the relationship has kind of shifted in the sense that I feel like I’m starting to become the parent in the relationship. But my mom always had this really good way of balancing.

Like Misty, several adult daughters mentioned they became closer to their mothers as they aged. Some also noted that becoming a mother shifted the relational dynamics with their mothers, as they compared parenting approaches and gained a deeper understanding of some of the decisions, challenges, and joys that coincide with parenting. In many cases, women with current healthy relationship patterns with their mothers felt closer to them when they reflected on their childhood experiences because they were better able to contextualize their mother’s choices. Bessie, a 33-year-old African American mother with one child, shared,

I was the only girl and I was the youngest. I was close to my mom because my older brothers would go off and play football and stuff. My grandparents had my mom in their 40’s and she was born in the 1950’s. She had a really old school, traditional upbringing. So my mom was super strict, and I started to realize that as I went to middle school and talked to my peers. We [brothers] would all get whooped if one of us got in trouble. So she was pretty strict, but she was also very easy to talk to.

Like with Bessie, a primary element that seemed to coincide with healthy and close relationship dynamics now involved the extent to which the women felt able to talk with their mothers (as children and as adults). In most cases (n = 13), the women received punitive consequences as a child and chose to shift these patterns as mothers; however, they were able to communicate with their mothers about mundane and significant life decisions, and they believed their mothers had their best interest at heart. Fannie, a 40-year-old African American mother with two children, mused,

Authentic. I have a really good relationship with my mother now, but when I was a kid! Woo. She raised me to be strong, independent, and free thinking. Doesn’t translate well when you’re strong, independent, and free thinking as a child. Right? Because that means you don’t always agree. It means you have a mind of your own. It means you do what you want to sometimes. That was not how my mother was raised. She was raised as a person who followed her mother. It wasn’t a democracy with my grandmother. She pretended like it was, but it really wasn’t. So with my mom, I don’t think she realized at the time that she wasn’t raising us in a democracy. And now, as an adult, you reflect and talk about it, and my mother will be like, “That’s not how that went.” But it was. We had a discussion recently about the way she raised me. She said, “I’m not even sure if I raised you to be that person. You guided me into that as who you are.”

Overall, their narratives indicated they had healthy mother–daughter relationships because they had a sense of trust and connection with their mothers. The women felt like their mothers actively listened to their thoughts and beliefs and they acknowledged their feelings, including reflections and feelings about how they were raised in comparison to how they were raising their children. A few adult daughters mentioned that it was important to them that their mothers were non-judgmental, even when they experienced life stressors (e.g., divorce, children getting in trouble at school, or unemployment). Furthermore, it seemed critical for these women—the adult daughters and their mothers—to be able to see each other as whole people who were doing their best with the life knowledge and experiences they had at their disposal. This shared empathy and understanding allowed daughters and mothers to contextualize and move forward from difficult childhood experiences, and it fostered more appreciation for the family traditions, values, and positive memories that the women carried with them as they stepped into their roles as mothers.

## 6. Discussion

In the current study, we considered Black adult daughters’ perspectives on maternal disciplinary practices as an intergenerational component of parenting and mother–daughter attachment. Consistent with intergenerational narrative frameworks ([37]; [55]; [56]), we found that Black adult daughters simultaneously came to understand themselves and their mothers in news ways as they made sense of their familial past. Our findings also demonstrated how Black adult daughters used maternal disciplinary narratives to construct a sense of their own motivations and goals as mothers. While exploratory, we believe our findings enrich attachment theory ([9]; [20]; [28]; [79]) and expand knowledge on the developmental implications of disciplinary approaches as a tethering point to broader family dynamics ([1]; [31]; [39]).

The first main finding is that Black adult daughters in the sample, as well as their mothers, used a range of disciplinary practices and approaches with their children. These findings are consistent with previous work documenting disciplinary heterogeneity among Black mothers ([49]; [57]; [70]; [89]). We also found indications that adult daughters who experienced spanking or harsh physical punishment tended to recall psychologically or emotionally harmful disciplinary practices from their mothers, as well. For instance, some of their mothers frequently yelled when they were upset, used the silent treatment or ignored their daughters, or tried to invoke guilt or shame in response to their daughter’s misbehavior. In recalling these events, the women talked about how these types of disciplinary practices made them feel momentarily unseen or unwanted, especially when they were a common and consistent occurrence in the home. These results are aligned with related research intimating that physical discipline patterns tend to correspond with emotionally immature responses from parents who do not have the adequate tools, skills, or resources to remain calm and connected to their children during moments of misbehavior or parenting stress ([2]; [4]; [36]).

For some adult daughters, spanking and physical punishment were a very infrequent form of discipline, and they more often received other forms of restrictions from their mothers instead (e.g., grounded from leaving the home or taking away toys or access to television). Although infrequent, it was still a salient memory that the women recalled, and only one mother in the sample still mentioned using physical punishment or spanking as a discipline practice with their children. Thus, although punitive consequences were the most frequent category for the type of discipline the adult daughters experienced during their childhood, most women in the sample recognized the harmful emotional, psychological, and physical consequences of such treatment ([4]; [36]) and they were choosing to use other methods with their own children. The second most frequent category involved logical consequences ([52]), in which their mothers imposed a consequence that was directly connected to their daughter’s perceived misbehavior. For instance, one adult daughter talked about how her mother took her phone away for a month after she was caught talking on the phone after her bedtime curfew. In most cases, logical consequences corresponded with mothers talking with their daughters about the misbehavior and the consequences of their actions ([74]). In recalling these childhood events as adult women with children of their own, they were better able to articulate the values and lessons their mothers were trying to impart as compared to mothers who recalled physical punishment as their mothers’ primary discipline method.

We were not able to locate other studies that considered natural consequences as a disciplinary approach within Black families. To date, most studies have focused on punitive or harsh disciplinary practices ([1]; [39]; [88]) or cross-racial and cross-cultural comparisons in parenting discipline ([6]; [17]; [23]; [24]; [28]; [31]). While only a couple of adult daughters in the sample mentioned natural consequences as their mother’s main discipline method, they suggested that this approach helped them learn that they were responsible for their own choices and they felt more equipped to think through the types of consequences that could follow from their actions. The relative infrequency of natural consequences as a main discipline category compared to punitive consequences among the women’s mothers in the sample is unfortunate, given established evidence (with mostly White samples of children) that natural consequences can contribute to a range of positive well-being outcomes among youth ([12]; [50]) when accompanied by secure parental attachment and supportive school and community environments.

Still, the women’s narratives were consistent with related work, noting that natural consequences can be effective in allowing children the opportunity to simply learn from their mistakes without the oversight or imposition of adult authority. Scholars should continue to examine the use of natural consequences among Black parents and youth, especially in the context of racism-related stress among Black families ([62]; [69]; [78]). Specifically, Black mothers may believe they are doing their children a disservice if they do not impose punitive or logical consequences to correct perceived misbehaviors early on, because Black people receive unfair and harsh treatment in U.S. society due to antiblack racism ([32]; [61]; [67]). Thus, creating a racially just and safe society for Black families may be a necessary antecedent to encouraging the more widespread use of natural consequences as a disciplinary approach.

The second main finding involved the women’s discussions of how their mother’s disciplinary strategies informed their current disciplinary practices as parents. Namely, shifting their disciplinary practices was the most common theme in the sample. Most mothers who experienced punitive discipline discussed the intentional shifts they had made with their own children; this was especially true for mothers who experienced physical harm, verbal abuse, or emotional manipulation. Most of these women thought that there were more effective methods of discipline that would teach their children long-term accountability and emotion regulation. Their primary focus on communication and connection parallels the literature ([46]; [68]; [80]), in the sense that fostering a sense of safety and high levels of support (even in moments of disciplinary correction) facilitates better social ([10]) and emotional ([85]) outcomes among youth. While less of this research focuses on Black children and adolescents, specifically ([79]), we know that family relationship quality promotes social competence ([87]) and positive mental health among Black youth ([62]; [71]).

Thus, the young women’s focus on deviating from control-based parenting strategies (e.g., spanking and yelling) to logical or natural consequences will likely have positive benefits for their children in the future. Conversely, mothers who experienced a combination of logical and natural consequences from their mothers tended to use a mix or most of the same discipline practices as their mothers. A key element that seemed to facilitate this continuity involved the level of open communication and trust that was maintained in the relationship. In trying to substantiate this finding with related literature, we were unable to locate similar studies that focused on intergenerational disciplinary practices. Instead, we will again point to analogous attachment scholarship that attests to the long-term and positive impact of having parent–child relationships that are characterized by trust, connection, openness, emotional expressivity, healthy boundaries, and empathy ([20]; [30]; [90]; [80]; [85]). In relation to intergenerational disciplinary continuities, our findings suggest that mothers who received this type of support might have felt a stronger desire to model similar parenting practices and approaches as their mothers.

Lastly, our third main finding involved drawing exploratory connections between adult daughters’ prior disciplinary experiences and their current relationships with their mothers. Although we used different typologies (i.e., healthy, progressing, and strained), the women’s descriptions of their current relationships with their mothers aligned well with the extant literature on attachment styles (i.e., secure, anxious, avoidant, and disorganized attachment; [5]). Women who did not receive adequate support, validation, and affection from their mothers throughout childhood revealed a higher likelihood of having a strained or progressing relationship dynamic with them in adulthood. In talking about their mother–daughter relationship dynamic, women in the strained and progressing category described characteristics of anxious, avoidant, or disorganized attachment patterns, such as having to avoid emotional intimacy or closeness with their mothers or limiting their interactions with them to maintain healthy personal boundaries. Although discipline is not the only factor that informs mother–daughter relationships, the ways that parents choose to engage their children in moments of stress and conflict during childhood likely has a significant influence on the overall communication and connection patterns within the relationship well into adulthood ([2]; [9]; [42]).

Finally, it is important to note that women’s punitive discipline experiences during childhood were not directly linked to a strained or progressing relationship dynamic with their mothers in adulthood. Instead, we found contextual nuances in so far as mothers who discussed more cognitive and affective understanding of their mother’s strategies evidenced more generational continuity in their disciplinary practices and more secure attachment patterns with their mothers. Similarly, mothers who experienced punitive discipline during their childhood, but also received accountability from their mothers in adulthood, seemed to evidence a sense of secure attachment and a healthy relationship with their mothers. Thus, our findings imply that sensitive and emotionally responsive parenting can play a positive role in mother–daughter relationship dynamics, even if they take years beyond childhood to develop.

## 7. Limitations and Areas for Future Research

Despite the strengths of the present study, it also has several limitations. First, the data were retrospective and involved momentary snapshots of the women’s childhood experiences. This precluded us from drawing any causal conclusions from our findings, even though it could be argued that the most saliently recalled moments were also the memories that had the strongest or most enduring impact when daughters thought about their mother’s discipline. Given evidence that individuals show superior memory accuracy for events they detail as emotionally negative ([22]), this saliency effect might have been stronger for adult daughters who reflected on disciplinary practices they considered harmful. Thus, a more comprehensive assessment of maternal discipline through additional interview questions, as well as triangulation data from multiple sources (e.g., mothers), for example, could offer more nuanced insight into Black women’s family dynamics in future studies. Second, our focus on maternal discipline figures may have masked the interactional nature of other disciplinary figures within their lives. When we inquired about their childhood experiences, most women also mentioned fathers, grandparents, and, to a lesser extent, extended family members like aunts and uncles. Thus, our primary focus on mothers might have obfuscated other significant figures who served as primary disciplinarians. Third, the women provided varying levels of depth in their responses about discipline. The interview findings could have been complemented by survey data that included a range of potential practices and general assessments of frequency as well as adult daughters’ stress appraisal of such events and their coping responses. Measures of coping, specifically, might have offered a unique opportunity to analyze how a daughter’s responses to their mother’s disciplinary practices may have informed the strength and connection in their relationship over time.

## 8. Conclusions

While discipline is only one aspect of parenting, it is a significant correlate of other parent–child relational dynamics, including connection, communication, and attachment. In the current study, we offered additional empirical evidence on how patterns of discipline and attachment can inform Black mother–daughter relationships well into adulthood. One implication of this work is that it reminds us to consider the importance of intergenerational socialization among Black families, including how Black adult daughters narrate and recall what they experienced in childhood. This remains a critically understudied area in thinking about how mothers equip their daughters with the cognitive and emotional tools necessary to navigate life experiences, in so far as Black daughters perceive their mothers as a secure base of support and guidance—even when they have made a mistake or violated a family, school, or community norm. A second important implication of our study is thinking about how to promote the types of safe and nurturing disciplinary practices that are essential for healthy growth and development. The prevalence of the ‘shift’ category within the current study suggests that Black mothers recognize the importance of diverging from punitive and/or physically harmful disciplinary practices that may do more harm than good. From an attachment perspective, these types of shifts will likely improve Black women and girls’ connections with their mothers across the lifespan.

## Figures and Tables

**Table 1 behavsci-15-00887-t001:** Mothers’ demographic information and discipline categorizations.

Pseudonym	Ethnicity	Age	Education	Children’s Ages	Discipline Recalled as a Child	Disciplinary Practices as a Mother	Current Relationship with Mother
Althea	African American	32	Graduate degree	3, 3	None	Shift	Progressing
Alyce	Biracial	41	Graduate degree	4	Punitive	Shift	Strained
Angela	African American	41	Graduate degree	5, 9, 12	Punitive Natural	Shift	Strained
Bessie	African American	33	Graduate degree	1	Punitive	Shift	Healthy
Charlie	African American	45	Graduate degree	8, 15, 17	Punitive Logical Natural	Continuity	Healthy
Claudette	African American	41	Graduate degree	21, 13, 13	Punitive Logical Natural	Continuity	Healthy
Daisy	African American	40	Bachelor’s degree	2.5	Punitive Logical	Mix	Healthy
Daisy	African American	39	Some college	2.5	Punitive	Shift	Strained
Deja	Caribbean	36	Graduate degree	9, 11	Punitive Logical	Mix	Progressing
Fannie	African American	40	Bachelor’s degree	3, 4	Punitive	Shift	Healthy
Florynce	African American	33	Bachelor’s degree	2.5	Punitive Logical Natural	Continuity	Healthy
Gwendolyn	African American	27	Bachelor’s degree	6, 3	Natural Logical	Shift	Progressing
Harriet	African American	33	Bachelor’s degree	3, 1	Punitive Logical	Shift	Healthy
Henrietta	African American	33	Some college	13, 11, 9, 7, 4, 11 mo	Natural Logical	Mix	Progressing
Ida	South African	43	Graduate degree	20, 16	Punitive Logical	Shift	Strained
Imani	African American	32	Trade school	11, 15	Punitive Logical	Continuity	Progressing
Jada	Ghanaian	26	Graduate degree	6, 6 mo	Punitive Logical	Continuity	Healthy
MaeJae	African American	33	Graduate degree	2	Punitive	Shift	Progressing
Mariana	African American	missing	missing	missing	Punitive Logical	Shift	Progressing
Mary	Biracial	33	Graduate degree	10 mo	Punitive	Shift	Healthy
Maya	African American	46	Graduate degree	17	Punitive Logical	Shift	Strained
Miriam	African American	27	Bachelor’s degree	12, 10, 8	Punitive Natural	Shift	Progressing
Misty	African	47	Graduate degree	19, 18	Punitive Logical	Shift	Healthy
Nova	African American	34	Graduate degree	5, 14 mo, pregnant	Punitive Logical	Mix	Healthy
Phyllis	Black American	44	Graduate degree	14, 4	Punitive	Shift	Healthy
Rosetta	Multiracial	28	Graduate degree	5, 3, 3 mo	Punitive	Shift	Strained
Shanice	African American	35	Graduate degree	1, pregnant	Punitive Logical	Mix	Healthy
Shirlee	African American	31	Graduate degree	8, 14 weeks	Punitive	Shift	Progressing
Soujorna	Biracial	43	Bachelor’s degree	14, 12	Punitive	Mix	Strained
Sylvia	African American	34	Trade school	8	Punitive Logical	Shift	Healthy
Violet	African American	36	Bachelor’s degree	4	Punitive	Shift	Progressing

Note. Ages of children—mo = months. For the current disciplinary practices as a mother, there were three main categories: continuity, mix, and shift. For the current relationship with mother column, there were also three main categories: strained, progressing, and healthy.

**Table 2 behavsci-15-00887-t002:** Discipline categories and example excerpts.

Type of Discipline	Excerpts
Punitive (*n =* 23, 74%) Consequences rooted in negative reinforcement; consequences are given by an adult in response to a child’s perceived misbehavior; punishment may not be connected to the perceived misbehavior	“She would put me on punishment, whoop me, or she would give me the silent treatment. Restrictions, no TV, you can’t go out, stuff like that. Mostly it was no TV. I was never really a social butterfly, so her saying, “You can’t leave the house,” didn’t bother me at all.” (Imani, 34, African American) “I think probably the central theme, we come from a culture where respect for elders is very much kind of like a pillar culturally. So we were kids that got spanked, but not often. I think it was so rare that I can remember every time that I ever got spanked because it was so rare. I think the last time I ever got spanked I might’ve been like eight years old. But pretty much, once you started heading into adolescence, spanking was no longer a part of how she parented. It was more so restrictions in terms of where you could go. So grounding you—you are not allowed to participate in certain events if you were misbehaving or something like that. The physical discipline was more so concentrated on the earlier part of childhood.” (Nova, 43, South African) “I remember I used to ask for whoopings because my mom was verbally abusive. She would just yell. It makes me so anxious when people yell. I feel like my insides are shaking, I can’t take it. So I just used to ask her, “Can you just whoop me and get this over with?” She kind of talked to me and treated me poorly. There was just no respect there. I remember her at least one time calling me a bitch, and I was like, “Oh my gosh, I got to get the fuck out of here.” She was so hateful sometimes. And I wasn’t a bad kid. I mean, I did typical stuff but I wasn’t doing nearly as much as I could have. She would say anything, it would sound so harsh. I can just remember the way it felt, it just felt like there was just hatred in there. Like, “I don’t understand why she doesn’t like me.”” (Shirlee, 32, African American)
Logical (*n =* 18, 58%) Consequences imposed by an adult that were tied to a perceived rule violation by a child; could be punitive in nature; often involved communication about misbehavior and the consequence	“Discipline was just ongoing. It was telling you what you needed to do and reminding you of what you needed to do. If something came up, it was discussed. If I didn’t do something I was supposed to do, then it was like, “Okay, you didn’t do this. How come you didn’t do it?” It was more like teaching, more conversational.” (Miriam, 32, Black American) “We had things taken away, we couldn’t watch TV, we weren’t able to go out with friends. Lots of conversations. Kind of explain to me why you felt like that was a good idea, type of thing. We got a lot of “What do you think the consequence should be for this?”” (Claudette, 36, Caribbean) “It was authoritarian and like, “We make the rules, you follow them, and if you don’t, we’re going to find out about it, and do something.” So that was tough, right? It doesn’t really teach the problem-solving for the relationships between the siblings and things like that. Or like, one year, I found the Christmas gifts and then she took my stuff away.” (Gwendolyn, 33, African American)
Natural (*n =* 3, 10%) Consequences that were a direct result of a child’s behavior and were not imposed by an adult	“In discipline, she’s always giving us the benefit of the doubt. Like more than she should have, I think. And when I got pregnant at 18, she just instantly did everything she could to make sure that I was fine and felt supportive. And no judgment at all, honestly. And from there, too, because I had a period in my adolescence where I was like really acting a fool. Not disrespectful. We didn’t have a contentious relationship. I was just quietly doing whatever I wanted. But still being nice to my mom. Just not doing what I was supposed to do. And so it wasn’t that surprising that I got pregnant, I guess. But from there because I had these secrets like that, it put a divide between our closeness, I feel, because I was double life’ing it in high school.” (Florynce, 27, African American)
Multiple Categories (*n =* 15, 48%)	“When I was younger, she was whooping us. But as we got older, she would have talks with us, conversations with us and tell us right and wrong and how it was wrong. She was such a great parent. As you hear, I didn’t grow up with my Dad. They got a divorce when I was an infant. It was just her and three kids, and I think that she did a great job. We went through the homelessness and we had to stay together. We had to stay secretive so we wouldn’t end up in foster care, right? And so it made us significantly closer. Us three, we are super, super close. We’re both adults now and my oldest brother is doing great.” (Bessie, 32, African American, Punitive + Logical) “My mom was very into natural consequences. She would say or do something like, hey tie your shoes or you’re going to fall. In the end, when you fall, you fall. Just life teaches you fast. She didn’t really yell or anything like that, but it was very much—like a nag. I don’t want to call it authoritative, because it wasn’t but it was just like a nag. Like, did you do the dishes? You didn’t do the dishes. Okay, now you got to do the dishes for two weeks.” (Henrietta, 27, African American, Logical + Natural)

Note. Since many mothers offered examples that included multiple types of discipline, we categorized all noted behaviors within an excerpt and documented the multiple experiences (e.g., punitive and logical).

## Data Availability

Abridged versions of the interview protocol are available upon request to interested researchers.

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
