# Peer review of "An Intergenerational Exploration of Discipline, Attachment, and Black Mother–Daughter Relationships Across the Lifespan"

_behavsci, 2025, doi:10.3390/bs15070887_

Round 1

Reviewer 1 Report

Comments and Suggestions for Authors

This article is focused on applying an intergenerational narrative lens to Black motherhood, specifically disciplinary strategies. I think this article will make a positive contribution to the field and focuses on a group that has been historically stereotyped or excluded completely from research. Strengths of this article are the well-explained theoretical foundation, high level of writing quality and the focus on allowing Black participants to tell their own stories through narrative approaches. I have a few suggestions to further strengthen the paper. 

Methods: I think the methods would be strengthened by having a positionality statement including identities that the research team who conducted the thematic analysis hold and how that may have impacted interpretation of the data. I appreciate that the race and ethnicities of the study team were included and I think it would be helpful to include more information including status as a parent, or SES, for example, to help the reader understand the researchers’ positionalities.

Discussion: I think the first paragraph of the discussion (through line 714) might be better suited for the introduction as it outlines a significant gap in research on discipline strategies of Black mothers.

Author Response

We have attached our revision letter.

Reviewer 2 Report

Comments and Suggestions for Authors

The article used qualitative methodology to understand Black mothers' disciplinary strategies.  More specifically, the authors examined how Black adult daughters make sense of the disciplinary strategies they received in childhood, how their meaning-making of these experiences informs the discipline strategies they use with their own children, and how it informs their current relationships with their mothers. I read the article with great interest and enthusiasm! The movement from a deficit-oriented approach to studying discipline in Black families (specifically for Black mothers) is commendable. This work also fills many important gaps, such as understanding how (adult) children view their parent’s discipline and potentially examining long-term implications of discipline in Black families. Despite the article's strengths, I had concerns about the protocol and the statements used to support some themes. Below are some of my specific concerns.

Greater clarity is needed about the specific questions that were asked. It would be helpful to present the questions in the order in which they were asked to the participants or, if possible, include the full protocol as supplementary materials. This would help clarify many questions about the order in which questions were asked and exactly what participants responded to.  

Some of the points discussed in RQ2 under the continuity section seem like they should have been addressed when discussing RQ1 because it focuses on the description of the parents’ discipline style rather than on whether or not the mothers would continue with that discipline style. As a reader, I was also looking for more direct quotes to support the continuity theme. As it is currently presented, I don’t see that theme as particularly strong.

  “…most mothers described eliminating or reducing the types of punitive discipline they experienced with their mothers during childhood but retaining many of the family values or traditions”. It's unclear why quotes supporting this statement would be considered mixed because family values and traditions are not discipline strategies.  

I think RQ3 is really important, but I don’t see strong evidence for an explicit connection between disciplinary strategies and current relationship status, especially in the progressing and strained descriptions.  The quotes provided for the progressing and strained categories are more of a description of the relationship without connection to discipline—perhaps the authors need to clarify the exploratory nature of this question further or reframe how the results are presented.
Specific comments:

 Is there information about the mothers' marital status? This information may help us further understand who these women are and to whom the findings may generalize. It also provides information about the context in which these women are now raising their children—this probably has implications for discipline.

The number for multiple categories in the text doesn’t match Table #2The first paragraph should be more focused---lots of really great information but some of it is not immediately relevant for building the foundation for the study discussed.

Author Response

We have attached our revision letter.

Round 2

Reviewer 2 Report

Comments and Suggestions for Authors

The authors made excellent revisions to the manuscript.  However, I found some inconsistencies in the description of the interview questions in the manuscript and the protocol in the supplementary materials for RQ3. 

Author Response

The authors made excellent revisions to the manuscript.  However, I found some inconsistencies in the description of the interview questions in the manuscript and the protocol in the supplementary materials for RQ3. 

Response: In the last revision, we condensed the full interview protocol to be on one page, so we'd left off a few probing questions. We included the probing questions in the revision so that the questions are consistent. We highlighted these changes in the manuscript and in the interview protocol appendix.

Please see the revised manuscript.